# Physical Activity in 6.5-Year-Old Children Born Extremely Preterm

**DOI:** 10.3390/jcm9103206

**Published:** 2020-10-04

**Authors:** Jenny Svedenkrans, Örjan Ekblom, Magnus Domellöf, Vineta Fellman, Mikael Norman, Kajsa Bohlin

**Affiliations:** 1Department of Clinical Science, Intervention and Technology, Division of Pediatrics, Karolinska Institutet, 141 52 Stockholm, Sweden; Mikael.norman@ki.se (M.N.); kajsa.bohlin@ki.se (K.B.); 2Department of Neonatal Medicine, Karolinska University Hospital, 141 86 Stockholm, Sweden; 3The Swedish School of Sports and Health Sciences, 114 86 Stockholm, Sweden; orjan.ekblom@gih.se; 4Department of Clinical Sciences, Pediatrics, Umeå University, 901 85 Umeå, Sweden; magnus.domellof@umu.se; 5Department of Clinical Sciences, Lund, Pediatrics, Lund University, 221 84 Lund, Sweden; vineta.fellman@med.lu.se; 6Children’s Hospital, University of Helsinki, 000 14 Helsinki, Finland

**Keywords:** infant, extremely premature, prospective studies, accelerometry, follow-up studies, brain injury, cardiovascular risk, neonatal sepsis, exercise

## Abstract

Physical activity (PA) can prevent cardiovascular diseases. Because of increased risks of impairments affecting motor activity, PA in children born preterm may differ from that in children born at term. In this prospective cohort study, we compared objectively measured PA in 71 children born extremely preterm (<27 weeks gestational age), to their 87 peers born at term, at 6.5 years of age. PA measured with accelerometer on the non-dominant wrist for 7 consecutive days was compared between index and control children and analyzed for associations to prenatal growth, major neonatal brain injury, bronchopulmonary dysplasia and neonatal septicemia, using ANOVA. Boys born extremely preterm spent on average 22 min less time per day in moderate to vigorous physical activity (MVPA) than control boys (95% CI: −8, −37). There was no difference in girls. Amongst children born extremely preterm, major neonatal brain injury was associated with 56 min less time in MVPA per day (95%CI: −88, −26). Subgroups of children born extremely preterm exhibit lower levels of physical activity which may be a contributory factor in the development of cardiovascular diseases as adults.

## 1. Introduction

The chance of survival for infants born in Sweden between 22 and 26 weeks gestational age (GA) increased from 70% to 77% between 2004–2007 and 2014–2017 [1]. Consequently, there are more survivors of preterm birth growing up, resulting in more people at risk of developing long-term consequences in later life [2,3,4,5,6,7,8,9]. It is well-known that inactivity in childhood is associated with health risks in adulthood [10,11] while physical activity (PA) is beneficial and can prevent major chronic diseases such as cardiovascular diseases and diabetes [12,13,14]. The knowledge about PA in children born extremely preterm is scarce and show conflicting results. Most studies used self-reported activity levels as a measure of PA, where most found lower levels [3,15,16,17], but some equal levels as term peers [18,19]. In the few studies using objectively measured PA, two found no differences [3,20] whereas one showed less PA in boys born before 32 weeks GA [21]. More studies are needed to clarify the association between preterm birth, physical activity and its association with long-term cardiovascular outcome.

Several neonatal morbidities could possibly affect PA in later life. Severe brain injuries caused by intraventricular hemorrhage (IVH), periventricular leukomalacia (PVL) or septicemia-induced cerebral inflammatory response [22] may affect motor function [23]. Bronchopulmonary dysplasia (BPD) and decreased lung function [3,16], as well as growth restriction and low birth weight may affect exercise capacity [24]. The correlation of perinatal morbidities and later PA has not been explored sufficiently, and more and larger studies are needed to further understand the relationship of extremely preterm birth and PA in later life.

We hypothesized that survivors of extremely preterm birth would exhibit lower levels of PA in childhood than peers of the same age born at term. Furthermore, we hypothesized that perinatal morbidities would significantly contribute to differences in PA levels. To test these hypotheses, we compared objectively measured PA between children born extremely preterm and term peers and correlated the outcomes of PA to perinatal morbidities.

## 2. Participants and Methods

### 2.1. Ethics

The study was approved by the Regional Ethics Review Board in Stockholm (no. 2010/520–31/2). Written informed consent was obtained from parents/guardians of all included children.

### 2.2. Study Cohort

The study subjects were collected from the EXtremly PREterm infants in Sweden Study cohort (EXPRESS). EXPRESS is a prospective population-based cohort, which included all live born infants with a GA less than 27 weeks, born by mothers residing in Sweden, from 1 April 2004 to 31 March 2007, Detailed characteristics of this cohort have been published previously [25,26,27]. All surviving children in the EXPRESS cohort were invited to a neurodevelopmental follow-up at 6.5 years of age ±3 months [28]. In addition, children born in three regions (Lund, Umeå and Stockholm; *n* = 250) were also invited to a comprehensive cardiovascular and respiratory assessment, including accelerometry to measure PA. Exclusion criteria were congenital cardiovascular or pulmonary malformations, ongoing chronic cardiovascular conditions or ongoing chronic lung disease except asthma or BPD. The Swedish Medical Birth Register was used to identify control children born at term who were matched on sex, date of birth, birth hospital, residential region and maternal country of birth. A detailed description of the cohort has been published previously [29] as well as blood pressures and heart and lung function in these children [30,31,32]. Written informed consent was obtained from parents or guardians before inclusion in the follow-up study. Inclusion flow chart is shown in Figure 1. Children born extremely preterm are referred to as “index” whereas term children are referred to as “controls”.

### 2.3. Measurement of Physical Activity

At the follow-up appointment, accelerometers (Actigraph GT3X+, Pensacola, FL, USA) were distributed by research nurses to all study subjects (index and controls). The accelerometer was applied on the non-dominant wrist by the nurse, and the participants and their parents were instructed to keep the accelerometer on for seven consecutive days. Wrist-worn accelerometers were chosen since they have been proven reliable and imply higher compliance in children [33,34]. Parents were given a diary and asked to fill out if the child became sick during the week. After the study period, accelerometers and diaries were sent by mail to the investigators for further analysis.

Data from the accelerometers were analyzed in Actilife (version 6.13.3, Actigraph, Pensacola, FL, USA). Triaxial vector magnitude data with a 60 seconds epoch were used. The limits for the activity levels sedentary (SED) and moderate to vigorous physical activity (MVPA) were chosen according to Chandler [35] (0–3660 counts/min = SED, 3661–9804 counts/min = light, >9804 counts/min = MVPA). Recordings from 7 am to 8 pm were used. To be considered a valid measurement, the child had to have worn the accelerometer for at least 4 days, at least 10 hours per day on average and not been sick for more than 50% of the days. Episodes of more than 60 min without counts but allowing spikes (>199 cpm) of up to 2 min were considered as non-wear time.

Since the cut-points used [35] were evaluated for 8–12 year old children, and our children were 6.5 years old, we chose to add a 10 level scale of intensity of PA. The scale was created using the highest counts per minute amongst the subjects divided into 10 groups of equally large ranges. The ranges used were 0–4627; 4628–9255; 9256–13,882; 13,883–18,510; 18,511–23,137; 23,138–27,762; 27,763–32,392; 32,393–37,019; 37,020–41,647 and 41,648–46,274 counts/min.

Our main outcome was average time per day spent in Moderate to Vigorous Physical activity (MVPA, minutes). We also calculated proportions of time spent in MVPA, in sedentary physical activity (SED) and in the 10 different levels of PA.

### 2.4. Descriptive Data for Birth, Perinatal Morbidities and Follow-Up

Birth characteristics and perinatal morbidities (Table 1) for the index children were collected prospectively within the EXPRESS study as described previously [25,27]. Information about birth characteristics for the control children was collected from the mother’s obstetric medical record at the time for follow-uHeight and weight of the children were measured at the follow-up visit. Clinical examinations and parental questionnaires were used for the diagnosis of cerebral palsy (CP).

To evaluate the representativeness of the included participants, data from the included index children regarding GA, birth weight (BW) and perinatal morbidities (IVH, PVL, necrotizing enterocolitis (NEC) and sepsis) were compared first, to the characteristics of the complete EXPRESS cohort [27], and second, to the children not producing valid accelerometer data (drop-outs). There were no statistically significant differences except that sepsis was more common in the included index children compared to the EXPRESS cohort (54.9% vs 41.0%, *p* < 0.05). In addition, index and control children were compared regarding characteristics at follow-up (height, weight, sick days), which were also tested for correlations to the outcomes of PA.

### 2.5. Statistics

IBM SPSS version 26 (IBM, Armonk, NY, USA) was used for statistical analysis. Descriptive statistics were presented as means and standard deviations (SD) for normal distributed variables, median and ranges for non-normal distributed variables and as counts and proportions (%) for categorical variables. Statistical comparisons of groups (index vs controls, included vs drop-outs and included vs the full EXPRESS cohort) were performed using independent t-test (continuous outcomes), chi^2^ (categorical outcomes) and Mann–Whitney U test (non-normal distributed outcomes) and presented with *p*-values and confidence intervals (CI). For calculation of confidence intervals for non-parametric tests, the Hodges–Lehman estimate was used.

Correlations of birth characteristics and perinatal morbidities and PA were analyzed using linear regression for continuous variables and Analysis of Variance (ANOVA) for categorical variables and adjusted analyses. All analyses were tested for robustness.

All cofactors and covariates were tested if affecting the outcome. Cofactors and covariates which significantly or close to significantly (*p* < 0.1) were included in the initial model, and non-significant factors were excluded in a step wise manner. The results were presented as means, confidence intervals, and *p*-values.

*p*-values < 0.05 were considered statistically significant. A total of 14 subjects (8 index, 6 controls) lacked information about sick days. Multiple imputation of data, using 10 different imputation datasets, was used to include subjects with missing data in the adjusted analyses including sick days.

## 3. Results

### 3.1. Study Cohort Characteristics

Of the 193 (94 index and 99 controls) children approached to participate in the study, 158 subjects produced a valid accelerometer recording (71 index and 87 controls). In 32 cases, the accelerometers came back without any recorded data and were considered as not used, 3 children were excluded due to invalid data (too short wear-time, *n* = 2, or too many sick-days, *n* = 1) (Figure 1). Most included study subject wore the accelerometer the whole study period (median 7 days, range 4−7). Birth characteristics, perinatal morbidities, and characteristics at follow-up of the included children are shown in Table 1.

### 3.2. Comparison of PA Levels in Index and Control Children

The average time per day in MVPA in the cohort was 96.5 min (SD 32.1 min, 95% CI: 91.4, 101.5), the average proportion of time spent in MVPA was 12.8% (SD 4.2, 95% CI: 12.2, 13.5), and the average proportion of time in SED was 52.3% (SD 8.4, 95% CI: 51.1, 53.4). Adjusted comparisons of the levels of PA between index and control children revealed that there was a significant difference of time spent in MVPA of -25 min (Table 2, Figure 2) and -3.2 percentage units (Table 2) for boys born extremely preterm compared to boys born at term. Extremely preterm boys also spent 4.5 percentage units more time in SED compared with boys born at term (Table 2). There was no difference in the levels of PA when comparing all index children with all control children or when comparing index girls with control girls (Table 2).

Analysis of time spent in the 10 equally distributed count levels of PA, created to compare PA regardless of cut-points created by Chandler [35], revealed no statistically significant differences between all index children and all control children. However, when stratifying on sex, a significant difference was seen in boys with more time spent in the lowest activity levels for index boys and more time spent in the higher activity levels for control boys (Appendix A).

### 3.3. Associations Between Perinatal Risk Factors and PA

Perinatal morbidities, birth characteristics and characteristics at follow-up (Table 1) were tested for associations with levels of PA. In ANOVA analyses, adjusted for sex and co-variates and co-factors that significantly affected the outcome of PA, severe neonatal brain injury (IVH ≥ grade 3 and/or PVL) was shown to be the strongest predictor of less time spent in MVPA. Children with severe brain injury spent in average 56 min less time (95% CI: −87.5, −25.5 min) in MVPA per day compared with children without severe brain injury (Table 3, Figure 2). Two out of 9 children with severe neonatal brain injury and one without severe brain injury were diagnosed with CP at follow-up, and all were regarded as mild. A history of blood culture verified sepsis was found to be correlated to more time spent in MVPA with a mean difference of 20 minutes per day (95% CI: 6, 34 min; Table 3, Figure 2). There was also an interactive effect of sex and severe neonatal brain injury, which was mainly related to the one girl with severe neonatal brain injury who exhibited low levels of PA. There was no correlation between GA, birth weight standard deviation score (BWSDS), male sex or severe BPD to time spent in MVPA in the children born extremely preterm in adjusted analyses. The correlations of predictors to the proportion of time spent in MVPA were similar to the average time spent in MVPA (Table 3).

Severe neonatal brain injury (IVH ≥ grade 3 and/or PVL) predicted more time in SED (mean 11, 95% CI: 0.02, 0.19) whereas no other factors were significantly associated to SED in adjusted analyses (Table 3).

## 4. Discussion

This is the first study to objectively measure physical activity in early school age, only focusing on children born extremely preterm (<27 weeks GA). In addition, it is the first study to correlate perinatal risk factors and morbidities to the outcome of physical activity. We show that extremely preterm boys are less physically active at 6.5 years of age than term boys and that severe neonatal brain injury is the strongest predictor of less PA among the index children. The difference in time in MVPA between index boys and control boys is larger than, for example, the difference seen between overweight and normal-weight children at the same age [36]. This reduction in PA may theoretically, given the increased risk of cardiovascular diseases in adults born preterm and the preventive effect of physical activity, be a link in the development of cardiovascular diseases later in life [37,38,39].

The strengths of this study are the relatively large number of extremely preterm born children included and the possibility to assess the correlations to perinatal morbidities. Physical activity was measured objectively using accelerometers for 4–7 days, 10 h per day on average, giving the possibility to evaluate how active these children are in daily life. The comparison with the main cohort (EXPRESS) revealed no differences in morbidities or birth characteristics except that sepsis was more common among the included children. Including the whole EXPRESS cohort would have increased the power and may theoretically have enabled findings of smaller differences between index and control children or correlations between morbidities and PA. Nevertheless, the similarities between the study subjects and the full EXPRESS cohort indicate that the results might be generalized to the extremely preterm population in Sweden.

There are few studies measuring PA in extremely preterm children, and to the authors’ knowledge, only one measuring PA at the same age [21]. While the Millennium cohort included children with a GA up to 32 weeks [21], our cohort consisted of only extremely preterm children born at <27 weeks GA. Our data are concurrent, showing a decreased time in MVPA in preterm boys, but furthermore, we show the correlation to severe neonatal brain injury. The difference in boys in our data is also larger, with 20 min less time in MVPA per day compared to about 60 min per week [21]. This could be attributed to the difference in GA but also to the use of different cut-points and placement of accelerometer.

The design of the study does not allow to show whether these differences have any impact on future health or if increased PA would improve outcome or even be possible to achieve. However, previous research showing correlations between PA and cardiovascular diseases indicate that this could affect future health.

The most important limitation to the study is that there are no validated cut-points for wrist-worn accelerometers available for the specific age group analyzed. The cut-points used in the study were validated for 8–12 year old children [35] while our children were 6.5 years old. The actual time spent in MVPA may be slightly overestimated since arm movements decrease gradually with age and cut-points for younger children are higher [40]. In contrast, validation studies on preschoolers have shown that wrist-worn accelerometers have the possibility to imply higher specificity in deciding time in MVPA compared to hip-worn [40]. Nevertheless, specificity has not been compared to hip-worn accelerometers for the cut-points used in this study. Studies utilizing hip-worn accelerometers reported less time in MVPA than we did [3,20,21]. More time in MVPA and less time in SED could be caused by slightly low cut-points. However, Swedish 6.5-year-old children are either in their last year in pre-school or in school-preparing activities, where large proportions of the day consist of playing. Moderate activity is, according to the Children’s Activity Rating Scale (CARS), equal to a brisk walk [41], and in the Chandler study, playground playing was coded moderate to vigorous [35]. The children in the study may exhibit PA in these levels. In addition, the cut-points do not affect the comparison between index and controls or the correlation to birth characteristics or perinatal morbidities. Analysis of PA in a 10-level PA scale also secure that the observed difference between index and control boys is not due to the cut-points for PA. Another limitation to the study is that the index children were shorter and had lower weight at the time of the test. These differences were expected and had no relation to the outcomes of PA. Furthermore, adjustment for height did not change the results.

A third limitation to the study is that the controls and the index children were not perfectly matched. There were slightly more index children amongst the dropouts. Although there were no statistical differences in morbidities between dropouts and included children, there may still have been differences not represented in the information available. In the diaries from the dropouts, parents reported autism as a reason for the children not wanting to wear the accelerometer. Even though not mirrored in our data, the dropouts may exhibit systemic behavioral differences compared to the tested children.

A fourth limitation of the study is the lack of information about activities that prohibit physical activity. We have information about sick days in most children, which had a small effect on PA in the comparisons between index and control children. However, we do not have information about how many days the children spent in school during the measurement and whether this affected the level of PA or not.

Severe neonatal brain injury is the perinatal event that significantly predicts less time in MVPA at 6.5 years of age, accounting for a difference of 56 min in MVPA per day. IVH and PVL are correlated to the development of cerebral palsy (CP), which may explain lower levels of PA [23]. However, only 3 children in the study cohort had mild CP, none had moderate to severe CP, and only two of the children with severe brain injury had CP, implicating that the decreased PA level is only partly explained by CP. However, IVH and PVL as well as prematurity have been correlated to clumsiness and impaired motor function [42,43,44], which may prevent children from being physically active. Hence, this finding was expected and confirms our hypothesis.

Given that septicemia during the neonatal period may cause brain injury [22], we hypothesized that septicemia would have an impact on PA. We showed that blood culture verified sepsis was correlated to an average of 20 min more time in MVPA per day in our cohort. The correlation of sepsis to the outcomes of PA raises the question that increases in PA may be a sign of hyperactivity or a different movement pattern. The risk of ADHD increases with decreasing GA and intrauterine growth restriction is a risk factor for ADHD [45,46]. Sepsis has been correlated to impaired motor skills [43], and in a study on very low birth weight preschoolers, sepsis was the strongest predictor of minor neurological dysfunction, which was also related to hyperactivity [47]. Consequently, the relationship between sepsis and PA could be mediated by increased hyperactivity. However, before drawing any firm conclusion, the correlation must be further explored in more studies.

It is well-known that children born extremely preterm are at risk of developing a decreased lung function [3], which was previously shown in this cohort [32]. However, a correlation between lung function and PA has not been confirmed [3,20]. Similarly, we found no correlation of severe BPD in the neonatal period to PA in childhood in this study. Although eight of the studied index children lacked information about oxygen supplementation at 36 weeks, a power calculation shows that the sample size should have been enough to show a difference of 15 min in MVPA. However, BPD diagnosis does not always predict pulmonary outcome, which decreases the likelihood to show a correlation to PA [32,48,49]. Consequently, a correlation between lung function and PA cannot be excluded and would require measures of lung function at the time of accelerometry, which is not included in our present data.

The difference in PA is only seen in boys, not in girls. In general, girls are less affected by extremely preterm birth and have better outcomes [50]. Most likely, some of the included children would fulfill diagnostic criteria for developmental coordination disorder which has been shown to reduce time in MVPA in boys, but not in girls [51]. In addition, the morbidity that explained most of the difference in PA in the index children, severe brain injury, affected only one girl but eight boys of our study subjects. When excluding the boys with severe brain injury from the analysis, there is still a difference of 15 min per day in MVPA between index and control boys, which is close to significant (*p* = 0.053), indicating that the difference in PA is explained by more factors than severe brain injury. The combination of better outcome and lower proportion of severe brain injury could contribute to the lack of difference in PA found in girls.

## 5. Conclusions

Boys born extremely preterm show a significant reduction in PA compared to term born peers at 6.5 years of age. Complications in the neonatal period can be linked to physical activity in early school age; severe neonatal brain injury resulted in less moderate–vigorous physical activity per day. Whether these differences remain during childhood and into adulthood and if these lower levels are correlated to an increased risk of cardiovascular diseases needs to be further explored.

## Figures and Tables

**Figure 1 jcm-09-03206-f001:**
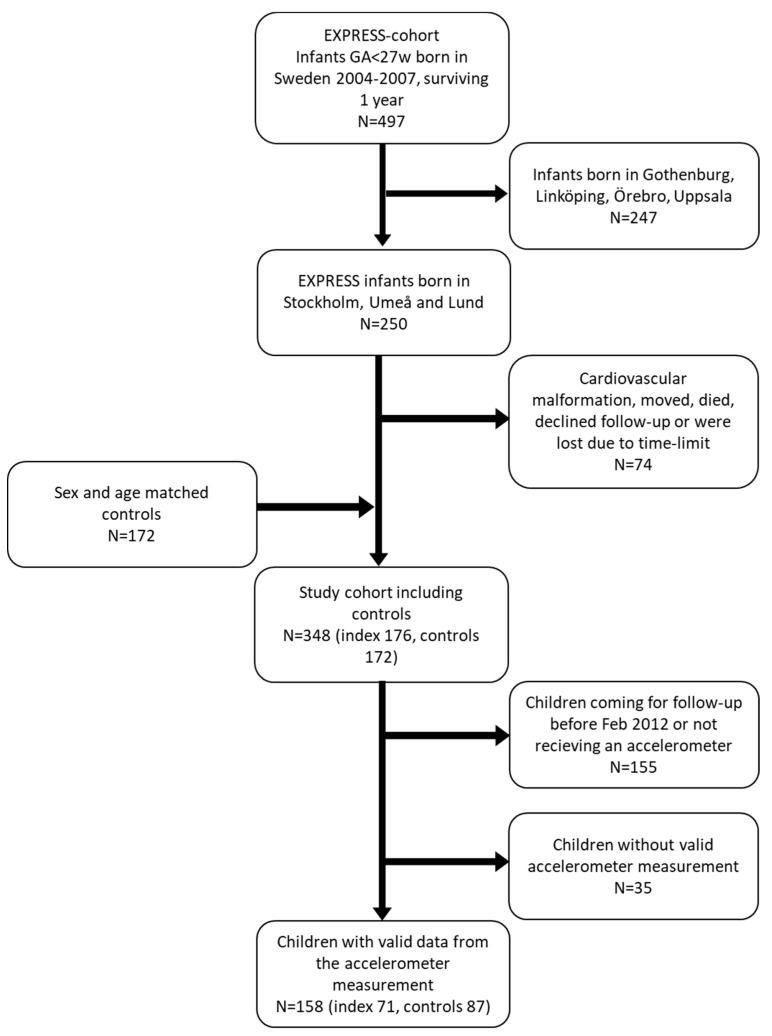
Inclusion of study subjects. The study subjects were recruited from the EXPRESS cohort. Matched controls were identified in the Medical Birth Register. The national follow-up study started in August 2010 and recruitment of participants to this study started in February 2012, After exclusion of subjects lost to time limit and drop-outs not producing valid data, results from 158 children were analyzed.

**Figure 2 jcm-09-03206-f002:**
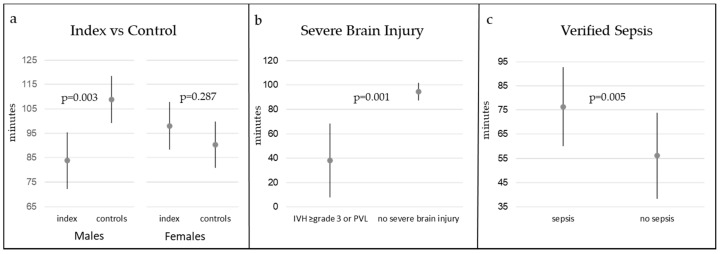
Average time in MVPA per day. (**a**) Comparison of average time spent in MVPA in index children and control children, stratified by sex. Error bars showing 95% confidence interval. (**b**) Comparison of time spent in MVPA between extremely preterm children with and without severe neonatal brain injury, error bars showing 95% confidence interval. (**c**) Comparison of time spent in MVPA between extremely preterm children with and without neonatal blood-culture verified sepsis. Error bars showing 95% confidence interval.

**Table 1 jcm-09-03206-t001:** Characteristics of study group.

Birth Characteristics	Index (*n* = 71)	Control (*n* = 87)	*p*
Male sex (*n*, %)	39 (54.9)	53 (60.9)	0.448
Gestational Age, weeks (mean, SD)	25.4 (1.0)	39.8 (1.2)	n/a
Birth Weight, g (mean, SD)	788 (160)	3625 (463)	n/a
Birth Weight Standard Deviation Score (mean, SD)	−0.77 (1.16)	0.18 (0.99)	n/a
**Perinatal Morbidities**			
ROP ≥ grade 3 (*n*, %)	18 (25.4%)	n/a	n/a
IVH grade ≥ 3 and/or PVL (*n*, %)	9 (12.7%)	n/a	n/a
Severe BPD (*n*/N *, %)	14/63 (22.2%)	n/a	n/a
Surgical NEC (*n*, %)	2 (2.8%)	n/a	n/a
Blood culture verified sepsis (*n*, %)	39 (54.9%)	n/a	n/a
**Characteristics at Follow-up**			
Height at test, cm (mean, SD)	118.7 (5.3)	124.1 (5.1)	<0.001
Weight at test, kg (mean, SD)	21.1 (3.8)	24.7 (4.2)	<0.001
Sick days during activity measurement (median, range)	0 (0−3)	0 (0−3)	0.395
mild CP (*n*/N *,%)	3/69 (4.3)	n/a	n/a

ROP—Retinopathy of Prematurity. IVH—Intraventricular hemorrhage, PVL—Periventricular Leukomalacia, BPD—Bronchopulmonary Dysplasia, NEC—Necrotizing Enterocolitis, CP—Cerebral Palsy * N=total number of children with information about BPD and mild CP.

**Table 2 jcm-09-03206-t002:** Comparison of levels of PA between index and control children.

Average Time in MVPA Per Day (min)	Index	Control	Difference	*p*
	Mean (95% CI)	Mean (95% CI)	Mean (95% CI)	
All	90.5 (82.6, 98.4)	100.8 (93.7, 107.8)	10.2 (−1.0, 21.4)	0.073
Males	83.8 (72.2, 95.4)	108.8 (99.1, 118.5)	24.9 (8.7, 41.1)	0.003
Females	98.0 (88.2, 107.9)	90.3 (80.8, 99.8)	−7.8 (−22.0, 6.5)	0.287
**Percentage of Time in MVPA**				
All	12.0 (11.0, 13.1)	13.4 (12.5, 14.3)	1.3 (−0.1, 2.8)	0.07
Males	11.2 (9.7, 12.7)	14.4 (13.1, 15.7)	3.2 (1.1, 5.3)	0.003
Females	13.0 (11.8, 14.2)	12.0 (10.8, 13.3)	−1.0 (−2.8, 0.9)	0.303
**Percentage of Time in SED**				
All	53.1 (51.2, 54.9)	51.4 (49.8, 53.0)	−1.6 (−4.2, 0.9)	0.211
Males	55.3 (52.6, 58.0)	50.8 (48.6, 53.0)	−4.5 (−8.2, −0.9)	0.018
Females	50.7 (48.5, 53.0)	52.5 (50.3, 54.6)	1.7 (−1.6, 5.0)	0.304

PA—Physical Activity, Index—children born extremely preterm, control—children born at term, MVPA—Moderate to Vigorous Physical Activity, SED—Sedentary Physical Activity. Analyses were adjusted for sex (when not stratified on sex), height and sick days, using ANOVA.

**Table 3 jcm-09-03206-t003:** Correlations of birth characteristics and perinatal morbidities to PA.

Average Time in MVPA Per Day (min)	Mean (95% CI)		*p*
	Yes	No	
Corrected ANOVA Model, R^2^ = 0.269			<0.001
male sex	80.2 (68.9, 91.5)	52.1 (23.0, 81.3)	0.078
Major brain injury	37.9 (7.7, 68.2)	94.4 (87.2, 101.7)	<0.001
Verified sepsis	76.3 (60.0, 92.7)	56.1 (38.3, 73.8)	0.005
interactive effect, sex and major brain injury			0.032
**Percentage of time in MVPA**			
Corrected ANOVA Model, R^2^ = 0.252			<0.001
male sex	10.7 (9.2, 12.2)	7.1 (3.3, 11.0)	0.087
Major brain injury	5.3 (1.3, 9.4)	12.5 (11.6, 13.5)	<0.001
Verified sepsis	10.2 (8.1, 12.4)	7.7 (5.3, 10.0)	0.007
interactive effect, sex and major brain injury			0.036
**Percentage of time in SED**			
Corrected ANOVA Model, R^2^= 0.176			0.011
male sex	55.8 (52.6, 58.9)	59.8 (51.7, 67.8)	0.361
Major brain injury	63.0 (54.6, 71.3)	52.5 (50.5, 54.5)	0.017
Verified sepsis	55.9 (51.3, 60.4)	59.7 (54.8, 64.6)	0.050
interactive effect, sex and major brain injury			0.092

PA—Physical Activity, MVPA—Moderate to Vigorous Physical Activity, SED—Sedentary Physical Activity.

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
