# Peer review of "Physical Activity in 6.5-Year-Old Children Born Extremely Preterm"

_jcm, 2020, doi:10.3390/jcm9103206_

Round 1

Reviewer 1 Report

This is a very well-written article outlining physical activity patterns in ex-preterm children of early school age. A novel finding of the study relates to the effect of severe neonatal brain injury on the amount of moderate to vigourous physical activity undertaken during habitual activity. The authours cite all important references and there is a detailed discussion of their results within the context of other studies. The authors describe that there are only a small number children with diagnosed CP. Thus there is valuable message that a group of children potentially have sub-clinical deficits in neuromotor function (e.g.developmental coordination disorder) affecting their participation in MVPA and survallience in this group could be of hightened importantance.

Author Response

Dear reviewer,

Thank you for your comments on our manuscript. No additional changes has been made to the manuscript, in accordance with the comments.

Kind Regards,

Jenny Svedenkrans

Reviewer 2 Report

The authors present a uniquely executed follow-up study on physical exercise at the age of 6 after preterm birth within a well-established collaboration network (EXPRESS) although not all centers were participating. This needs to be discussed with respect to the results obtained. The study is well-designed and the manuscript is well-written. The authors should elaborate on their finding that complications after preterm birth predispose for less activity as well as male gender-is there a connection or is this an independent observation. Can the results be confirmed if not a categorization but linear modelling is applied? The authors describe documentation of illness episodes during the observation period. They should as well take into account school attendance and other daily appointments were physical activity is prohibited. The authors have the data on lung function available in these children. As the difference is observed in males with a higher risk for lung function limitations, it would be worthwhile to work this out.

Author Response

Dear reviewer,

Thank you for your comments on our manuscript. We have addressed your comments below:

  • Not all centers participating in the follow-up
    • Thank you for highlighting this. As explained in the methods section, only three regions participated in the extended follow-up. A paragraph has been added to the discussion section to discuss this further.
    • "The comparison with the main cohort (EXPRESS) revealed no differences in morbidities or birth characteristics except that sepsis was more common among the included children. Including the whole EXPRESS cohort would have increased the power and may theoretically have enabled findings of smaller differences between index and control children or correlations between morbidities and PA. Nevertheless, the similarities between the study subjects and the full EXPRESS cohort, indicate that the results might be generalized to the extremely preterm population in Sweden." 
  • The authors should elaborate on their finding that complications after preterm birth predispose for less activity as well as male gender-is there a connection or is this an independent observation. 
    • Thank you for this comment. We have tried to clarify these findings. There is a difference in boys, that we think is partly explained by the fact that more boys had a severe brain injury. When comparing index boys with control boys, but excluding these with severe brain injury, the difference is close to significant (p=0.053). A paragraph har been added to the discussion regarding this:
    • "When excluding the boys with severe brain injury from the analysis, there is still a difference of 15 minutes per day in MVPA between index and control boys which is close to significant (p=0.053), indicating that the difference in PA is explained by more factors than severe brain injury. "
  •  Can the results be confirmed if not a categorization but linear modelling is applied? 
    • Yes, such analyses have been performed and the results have been confirmed. ANOVA has been chosen as statistical method, in discussion with statisticians, since the main aim was to compare outcome between groups.
  • The authors describe documentation of illness episodes during the observation period. They should as well take into account school attendance and other daily appointments were physical activity is prohibited.
    • This is a valuable comment. Unfortunately, we don't have information about school attendance and appointments when physical activity is prohibited. However, as stated in the discussion, school attendance at this age includes physical activity. A paragraph has been added to the discussion in order to further discuss this.
    • "A fourth limitation of the study is the lack of information about activities that prohibit physical activity. We have information about sick-days in most children, which had a small effect on PA in the comparisons between index and control children. However, we don´t have information about how many days the children spent in school during the measurement and whether this affected the level of PA or not.
  • The authors have the data on lung function available in these children. As the difference is observed in males with a higher risk for lung function limitations, it would be worthwhile to work this out
    • It is true that there is data on lung function available for this group of preterm children. The correlation of lung function data however includes too much information to be included in this work. The authors aim to explore that correlation in a future study.

We hope you find the comments and changes adequate for the paper to be published.

Kind Regards,

Jenny Svedenkrans

Reviewer 3 Report

The authors sought to quantify PA in a group of extreme preterm birth kids and term born kids. They found that preterm boys had significantly less time in MVPA than term boys, but no differences were present between girl groups. Likewise, they found that severe brain injury was the greatest predictor of less PA in the preterm group.

Overall, I found the study to be interesting and well-written. I have some comments/concerns that I hope the authors will consider.

  • Lines 118-119. It is not clear here that the authors are comparing the sub-sample of the entire EXPRESS cohort and the entire EXPRESS cohort.
  • Lines 179-181. This sentence could be edited for clarity. It’s not clear what is meant by “In adjusted ANOVA…” but perhaps that is a typo?
  • I find it particularly interesting that there was a lack of an effect of severe BPD on PA levels in the preterm groups. The authors have a relatively large sample of them in their group so this is surprising. Could the authors compute a power analysis? Were there differences between the severe BPD kids and the remaining preterm kids, i.e., a sub-analysis within only the preterm group?
  • Lines 205-206. I believe a reference is needed here to support this statement. I see that the authors are making a general statement about PA and CV disease, but I recommend being specific to preterms who may be at a greater risk. In the preterm group, though, this statement would be somewhat speculative as, to my knowledge, there are not longitudinal cardiac/CV data from childhood to adolescence to adulthood.
  • Line 225. Is “could” a better word than “may” here?
  • Lines 274-279. A sample size of 71 is quite large and having missing information on only 8 is really not many. It is my opinion that you have sufficient data to establish a link should one be present. The authors could compute their attained power to support their statement of an insufficient sample size.
  • Lines 280-282. This paragraph is highly specific and highly speculative. There are no data included in the paper to support these statements and I recommend omitting them.
  • Severe brain injury…more boys than girls? I may have missed it, but was this analysis specific to just the preterm boys or the entire preterm group?

Author Response

Dear reviewer,

Thank you for your valuable comments on our work. Please see the list below with our responses to your comments.

  • lines 118-119
    • Thank you for highlighting that this is unclear. The comparison has been made both to the EXPRESS cohort and to the drop-outs. We have tried to clarify this by changing the wording:
      "data from the included index children regarding GA, birth weight (BW) and perinatal morbidities (IVH, PVL, necrotizing enterocolitis (NEC) and sepsis) were compared first, to the characteristics of the complete EXPRESS cohort [27] and secondly, to the children not producing valid accelerometer data (drop-outs)."
  • lines 179-181
    • The sentence has been rewritten, which we hope has clarified what we have done: "In ANOVA analyses, adjusted for sex and co-variates and co-factors that significantly affected the outcome of PA, severe neonatal brain injury (IVH ≥grade 3 and/or PVL) was shown to be the strongest predictor of less time spent in MVPA."
  • Comment on no difference in PA in children with BPD.
    • We agree that this is interesting. We performed a power analysis showing that we had a sample size large enough to show a difference of 15 minutes in MVPA per day. The comment about sample size has been changed accordingly. Another possible explanation could be that BPD is a very blunt measure of lung function. Prevoius studies have shown that children born extremely preterm exhibit lower results in lung function testing, regardless of BPD. We have only tested for the correlation of BPD and PA within the extremely preterm group since there are to many other factors affecting the difference between index children and control children. When comparing the children with severe BPD  their time in MVPA was increased compared to children without BPD. However the difference was not significant.
  • lines 205-206
    • Thank you for this comment. It is a speculative statement and this could be claryfied. We have changed the sentence and added three references: "This reduction in PA may theoretically, given the increased risk of cardiovascular diseases in adults born preterm and the preventive effect of physical activity, be a link in the development of cardiovascular diseases later in life [37-39]. "
  • line 225
    • may has been changed to could
  • line 274-279
    • Thank you for observing this. See comment on sample size above. We have removed the comment about sample size and developed the discussion about lung function and extremley preterm birth. We also added a few references.
    • "Similarly, we found no correlation of severe BPD in the neonatal period to PA in childhood in this study. Although eight of the studied index children lacked information about oxygen supplementation at 36 weeks, a power calculation show that the sample size should have been enough to show a difference of 15 minutes in MVPA. However, BPD diagnosis does not always predict pulmonary outcome, which decreases the likelihood to show a correlation to PA [32, 48, 49]. Consequently, a correlation between lung function and PA cannot be excluded and would require measures of lung function at the time of accelerometry, which is not included in our present data. "
  • Lines 280-282
    • The paragraph has been omitted
  • Severe brain injury
    • We may not have understood the question completely, but have tried to give a comment as we understand it.
    • The analyses of correlations between major morbidities and the outcomes of PA were performed only within the index children. This was decided in discussion with our statisticians and was based on the fact that being an index child significantly increases the likelyhood of having BPD, major brain injury, sepsis etc. The correlation of severe brain injury to PA was found within the group of children born extremely preterm, regardless of sex. Sex was not shown to affect the outcome significantly. However, in the group of extremely preterm infants with severe brain injury, eight were male and one was female. Given that severe brain injury was the strongest predictor of lower levels of PA, and that this was more common among males, this may have been a contributing factor to the difference seen between preterm boys and boys born at term. A paragraph in the discussion has been changed in order to try to clarify this.
    • "In addition, the morbidity that explained most of the difference in PA in the index children, severe brain injury, affected only one girl but eight boys of our study subjects. When excluding the boys with severe brain injury from the analysis, there is still a difference of 15 minutes per day in MVPA between index and control boys which is close to significant (p=0.053), indicating that the difference in PA is explained by more factors than severe brain injury. 

We hope that you find that the changes and comments has been clarifying and sufficient for publication of our manuscript.

Kind Regards,

Jenny Svedenkrans, corresponding author.